# The Role of mTOR in Amyotrophic Lateral Sclerosis

**DOI:** 10.3390/biomedicines13040952

**Published:** 2025-04-13

**Authors:** José Augusto Nogueira-Machado, Fabiana Rocha-Silva, Nathalia Augusta Gomes

**Affiliations:** Stricto Sensu Postgraduate Program in Medicine/Biomedicine, Santa Casa de Belo Horizonte College, Belo Horizonte 30110-005, Brazil

**Keywords:** amyotrophic lateral sclerosis, ALS, mTOR, metabolic signaling, genetic biomarkers, neurodegenerative diseases, neuromuscular degeneration

## Abstract

**Background:** Amyotrophic lateral sclerosis (ALS) is a rare, progressive, and incurable disease characterized by muscle weakness and paralysis. Recent studies have explored a possible link between ALS pathophysiology and mTOR signaling. Recent reports have linked the accumulation of protein aggregates, dysfunctional mitochondria, and homeostasis to the development of ALS. mTOR plays a pivotal role in controlling autophagy and affecting energy metabolism, in addition to supporting neuronal growth, plasticity, and the balance between apoptosis and autophagy, all of which are important for homeostasis. **Aim:** This mini-review approaches the regulatory roles of mTOR signaling pathways, their interaction with other metabolic pathways, and their potential to modulate ALS progression. **Significance:** It discusses how these metabolic signaling pathways affect the neuromuscular junction, producing symptoms of muscle weakness and atrophy similar to those seen in patients with ALS. The discussion includes the concepts of neurocentric and peripheral and the possible connection between mTOR and neuromuscular dysfunction in ALS. **Conclusions:** It highlights the therapeutic potential of mTOR signaling and interconnections with other metabolic routes, making it a promising biomarker and therapeutic target for ALS.

## 1. Introduction

Amyotrophic lateral sclerosis (ALS) is an incurable disease characterized by progressive muscle weakness, resulting in paralysis [1,2]. There are two types of ALS: an idiopathic form known as sporadic ALS (sALS) and a genetically inherited type called familial ALS (fALS), which has over 30 identifiable subtypes of mutated genes. sALS develops epigenetic mutations that regulate the synthesis of mutated proteins, such as SOD1 and TARDBP [3,4]. The mechanisms associated with the onset and progression of this complex neurodegenerative disorder remain unknown. Nevertheless, the symptoms are similar among the several types and subtypes, and muscular weakness is the main one. Muscle weakness in both arms and legs, followed by respiratory muscle paralysis, is associated with neurodegenerative disorders. We must look into their genetics, proteomics, and environmental inducers to fully understand how they affect signaling pathways, especially those that control muscle function. Amyotrophic lateral sclerosis, one of the most significant motor neuron diseases, causes muscle degeneration and leads to paralysis [5]. The pathophysiological mechanisms of ALS are complex, involving intricate interactions among multiple factors and metabolic signaling. It is important to emphasize that frontotemporal dementia (FTD) shares significant pathological, genetic, and clinical similarities with ALS [6]. The pathophysiological processes underlying ALS are currently believed to be multifactorial. Four significant ALS-associated genes—C9orf72, SOD1, TARDBP (encoding the TDP-43 protein), and FUS—have multiple mutations and at least 40 other genes associated with fALS [7]. Studies have been performed on the role of innate immunity in ALS. Its pathophysiology involves pathogen-associated molecular patterns (PAMPs) such as High-mobility group box 1 protein (HMGB1), Receptor for advanced glycation end-products (RAGE), and Toll-like receptor 4 (TLR4) signaling axes, suggesting the perspective of the involvement of neuroinflammation in the disease [8]. On the other hand, it is also suggested that the downregulation of the IGF-1 (insulin-like growth factor 1) and GLP-1 (glucagon-like peptide-1) signaling pathways significantly affects the progression of ALS pathogenesis [5].

Figure 1 shows hypothetical metabolic interactions involving mTOR. Several growth factors, including IGF-1 and GLP-1, activate mTOR through the phosphoinositide signaling cascade involving Akt (PKB) signaling. The PI3K/Akt pathway can also activate mTOR. The complexity of the process becomes clear when PI3K/Akt/mTOR signaling enhances autophagy, and the activation of mTOR by growth factors can inhibit it, and apoptosis is activated. This imbalance, mTOR-induced, increases misfolded protein aggregation and deposition in the cytoplasm and endoplasmic reticulum.

The process could involve mutated SOD1, changes in antioxidative mechanisms leading to oxidative stress, and endoplasmic reticulum stress with impaired mitochondrial function. These processes are observed in ALS and other neurodegenerative disorders. The main symptom of ALS is progressive neuromuscular atrophy, and there is a proposed link between the dysregulation of IGF-1/GLP-1 signaling and ALS. By inhibiting the PI3K/Akt/mTOR and MAPK/ERK pathways, IGF-1 and GLP-1 prevent neuronal death resulting from amyloidogenesis, cerebral glucose deprivation, and neuroinflammation. Neurodegenerative disease progression accelerates in patients with IGF-1 resistance and GLP-1 deficiency [9,10,11,12,13]. The complexity of the process becomes evident when PI3K/Akt/mTOR signaling stimulates autophagy (Figure 1). At the same time, growth factors also activate mTOR, which subsequently inhibits it, and this inhibition leads to the activation of apoptosis. Despite the extensive literature on this topic, the relationship between signaling pathways related to mTOR activation and homeostasis remains unclear. Understanding these metabolic interactions in ALS is particularly complex. All interaction studies are open for discussion and study, including those on healthy individuals and those with specific pathologies.

It has been proposed that the IGF-1/Akt/mTOR signaling pathway improves muscle growth and controls the mTORC1-related protein synthesis pathways. These pathways are involved in both the hypertrophy and atrophy of the skeletal muscles, and they are less active in cases of muscular atrophy [14]. Therefore, the IGF-1/Akt/mTOR signaling pathway is believed to have a role in ALS. This review examines mTOR’s metabolic signaling interactions, which result in an imbalance between autophagy and apoptosis. This imbalance impacts cellular homeostasis. After extensive studies, mTOR could be a potential therapeutic target or biomarker for ALS and other neurodegenerative diseases that involve aggregation and misfolded proteins, among other conditions.

## 2. mTOR Structure and Function

mTOR belongs to the PI3K-related kinase (PIKK) family of protein kinases. There are two separate mTOR structures with catalytic units that control different cellular functions at the same time. The mTORC functionality is associated with neurodegenerative disorders. Three proteins comprise one of them, mTORC1: Deptor, PRAS40, GβL/mLST8, and the regulatory protein raptor (rapamycin-sensitive). The other structure, mTORC2, consists of the proteins GβL/mLST8, mSIN1, PRR5/Protor, Deptor, and Rictor (Figure 2). Studies have proposed that the mTORC2 protein Rictor is partially insensitive to rapamycin inhibition (Figure 2).

Growth factors, nutrition, and energy metabolism activate mTOR function. It regulates autophagy, contributes to lipids, proteins, and glucose synthesis, inhibits autophagy, controls ribosome function and mRNA translation, and enhances mitochondrial respiration. Therefore, the worsening of several diseases, such as cancer, diabetes, aging, and neurological disorders like ALS, has been associated with dysfunctional mTOR signaling [6,7,8,9,10,11]. Neuronal growth factors and other growth factors activate mTORC1. Decreased mTORC1 functionality can reduce memory retention and worsen muscle degeneration [6,7,8]. However, it can also stimulate lysosomal biogenesis, impacting homeostasis and influencing apoptosis, cellular development, division, proliferation, survival, and aging [11]. Understanding mTOR’s function is complex due to its multifaceted role. The same activators can trigger both pathways, yet each may have distinct characteristics (Figure 3).

mTOR inhibition triggers autophagy and proteolysis, which releases free amino acids for protein synthesis [12,13,14,15,16,17,18]. Both mTORC1 and mTORC2 have opposing effects on Akt. mTORC1 inhibits insulin/Akt signaling and also blocks mTORC2 activation. Conversely, the inhibition of mTORC1 using specific compounds, such as rapamycin, activates both Akt and mTORC2 [19,20,21,22]. Extracellular growth factors activate both mTORC1 and mTORC2 (Figure 2). However, the dysfunction or activation of mTORC1 leads to disturbances in proteostasis (impaired homeostasis) [23]. Another vital system for homeostasis is the action of AMPK (AMP-activated protein kinase), which may affect mTOR function. Both AMPK and mTORC function as energy sensors. Abundant nutrients activate mTOR, while calorie restriction activates AMPK. In this case, AMPK activates autophagy by the ubiquitin–proteasome system (UPS), a protein complex that induces proteolysis in misfolded proteins after being ubiquitinated and maintains homeostasis, detoxifies cells, and removes waste [24,25,26]. Despite the complexity of the mechanism, mTOR and AMPK are metabolic pathways that can be used to identify novel biomarkers or therapeutic targets. Studies have demonstrated that exercise, leucine, and IGF-1 activate mTORC1 signaling, promoting muscle hypertrophy in mice [21,27,28].

Mice given carbohydrate-restricted diets exhibited reduced mTOR activity, which had detrimental effects on memory and aging and increased lifespan, as shown by Robert et al. [29]. Mice lacking raptor, a component of mTOR, suffer premature death due to significant muscular atrophy and decreased body weight [29]. Rictor-knockout mice did not exhibit these results. It indicates that mTORC1, rather than mTORC2, is responsible for many of the essential functions of mTOR in skeletal muscle [21,29,30,31]. Muscular atrophy and weakness are common symptoms in all forms of ALS, suggesting a potential role for mTOR in the neurodegenerative pathophysiology. Interestingly, while the short-term, “in vivo” activation of mTORC1 signaling promotes muscle growth, long-term (chronic) activation can lead to significant muscle atrophy, body mass loss, and early death [29]. Persistent inhibition of autophagy and dysfunction in homeostasis within muscle tissue may be contributing factors. Grasping the functioning of mTORC1 across various signaling pathways related to insulin, proteins, lipids, amino acids, physical activity, and mechanical muscle force is essential. This perspective opens up a new way to look into treatments for various age-related diseases.

## 3. mTOR and ALS

ALS is a highly intricate neurological disease. In addition to the sporadic form, there are more than 30 subtypes of the genetic type, each with identifiable mutant genes. However, almost all studies concentrate on sporadic amyotrophic lateral sclerosis, with few reports on particular kinds of fALS. All types and subtypes of ALS share the same symptoms of atrophy and muscle weakness. Since mTOR is associated with muscular growth, a dysregulated mTOR function may play a role in the pathophysiology of ALS and possibly other neurodegenerative diseases [30].

Patients with familial ALS type 8 (ALS8) show elevated intracellular mTOR levels when compared to healthy controls. This increase in intracellular mTOR is suggested to be due to a possible dysfunction in the exocytosis process [31]. This finding has only been observed in cells from ALS8 patients and cannot be generalized to all types of familial ALS. The high intracellular mTOR content can inhibit autophagy and increase the presence of misfolded and aggregated proteins in the cytoplasm, as well as dysregulate the endoplasmic reticulum and alter enzymatic function, all of which could significantly affect both sporadic ALS and familial ALS. All these factors are directly or indirectly affected by mTOR.

Strong mTOR activation by nutrients and growth hormones, such as IGF-1 and insulin, promotes autophagy inhibition, possibly dysregulating homeostasis in ALS. It suggests a connection between the pathogenesis of ALS and the dysregulation of mTOR signaling and autophagy. Apart from oxidative stress, both sporadic amyotrophic lateral sclerosis and familial amyotrophic lateral sclerosis exhibit common features such as misfolded proteins, malfunctioning lysosomes, dysregulated mitochondria, changes in DNA and RNA activities, alterations in protein translation, and accumulation within the nucleus and cytoplasm. Mutant proteins are deleterious and affect cellular homeostasis. However, cells utilize various strategies to maintain cellular homeostasis, including well-balanced and activated autophagy. Gene mutations or epigenetic changes can cause ALS. These changes affect genomic and proteomic responses in different types of cells, which in turn affects the function of the neural plaque in ALS and other neurodegenerative diseases. This process involves mTOR-inhibiting autophagy. Dysfunction in mTOR signaling and astrocyte reactivity has been associated with neurodegenerative disorders and central nervous system damage [32,33,34,35].

Granatiero et al. [36] reported that in astrocytes with SOD1 mutations, the activation of mTOR is associated with the increased expression of the insulin-like growth factor 1 receptor (IGF1R). Elevated levels of the IGF1R activate the mTOR pathway and subsequently lead to the inhibition of autophagy. Interestingly, inhibiting the IGF1R-mTOR pathway has been shown to mitigate the harmful effects on motor neurons in the astrocytes of SOD1 mutant mice, indicating a potential connection between mTOR activity and motor neuron damage in ALS, highlighting mTOR’s complex and nuanced role [36].

However, the occasional conflicting reports about mTOR’s potential role in sALS highlight the need for further research. For instance, mTOR inhibition has been shown to both accelerate ALS progression and induce motor neuron degeneration in models with mutant SOD1. However, in other studies, similar mTOR inhibition in transgenic mice with elevated TDP-43 levels has led to protection rather than degeneration [33,34,35]. These findings underscore the complexity of mTOR’s function and emphasize the need for continued research. The quest for a definitive explanation is challenging.

*mTOR* is a natural autophagy inhibitor that alters the balance of autophagy and apoptosis. It could negatively influence homeostasis (Figure 2). Its inhibition could be a strategic therapeutic resource for controlling protein aggregation and accumulation inside cells from ALS patients and other neurodegenerative disorders. However, the possibility of negatively affecting cellular reactivity needs to be considered. Studies using rapamycin to inhibit mTOR have demonstrated its ability to reverse the autophagy inhibition it induces, leading to reduced internal cellular toxicity and improved homeostasis. Several factors, including hormones, intense physical activity, mechanical stress, and dietary intake, contribute to mTOR activation and muscle hypertrophy. Each factor impacts the mTOR signaling pathway, which is involved in muscle growth and interconnected with several other metabolic routes [37,38,39,40]. Mutations in valosin-containing protein (VCP) are associated with muscle weakness in mice. Dysfunction in mTOR and VCP may aggravate muscle weakness in amyotrophic lateral sclerosis (ALS), according to studies by James et al. [35]. It suggests that activated mTOR may play a role in the progression and exacerbation of ALS. Both the apoptotic and autophagic pathways in ALS are affected by mTOR signaling. An imbalance in these pathways can dysregulate homeostasis and cause disease [8,9,10] (Figure 3). Reports indicate that findings on mTOR inhibition are inconsistent, suggesting that results may change based on the type of ALS, specific gene mutations, and the metabolic signaling involved. In SOD1 mutants, mTOR inhibition worsens ALS, while similar mTOR inhibition in elevated TDP-43 levels protects against neuronal degeneration [41,42,43,44,45,46] despite the pathogenesis of ALS being associated with TDP-43 [47,48,49]. Amyotrophic lateral sclerosis and frontotemporal dementia have in common with each other their genetic and pathological characteristics. As the disorder progresses, there is an increase in neuronal degeneration, the formation of protein inclusions, chronic inflammation, and neurodegeneration.

Other signs observed in both ALS and FTD include the formation of misfolded TDP-43 and altered SOD1 or FUS. Furthermore, mitochondrial dysfunction, oxidative stress, protein imbalance, altered autophagy, ER stress-mediated cell death, stress granule formation, and altered RNA processing accelerate the disease’s progression [41,42,43]. Changes in homeostasis, cytotoxicity, and nucleocytoplasmic protein distribution correlate with TDP-43 aggregation and RNA dysregulation [39]. Multiple mutations in the TARDBP gene have been associated with ALS [50,51,52,53]. As a result, abnormal functions of TDP-43, including protein aggregation, relocation from the nucleus to the cytoplasm, and the accumulation of ubiquitinated and hyperphosphorylated TDP-43 in inclusion bodies, play a pivotal role in the disease’s pathophysiology.

These adverse effects are intensified in both sporadic and familial ALS [40]. Cytoplasmic inclusions of the transactive response (TAR) DNA-binding protein (TDP-43) are seen in about 90% of ALS patients. This intriguing phenomenon seems connected to reduced autophagy, potentially within an mTOR-activated system, which results in an increased accumulation of TDP-43′s C-terminal fragment. However, inhibiting mTOR pathways triggers autophagy, thereby regulating the cytoplasmic deposition of proteins. Rapamycin-induced mTOR inhibition stimulates autophagy, reducing TDP-43 (25 kDa fragment), regulating its accumulation, and returning it to its usual intracellular location [44,45,46]. These findings demonstrate the critical role of autophagy and the mTOR pathway in the pathophysiology of ALS.

Autophagy is a cellular process that clears protein aggregates, reduces caspase activation, and prevents apoptosis. This protective mechanism shields cells from the harmful effects of protein aggregates on their functions. Neurodegenerative diseases such as ALS have been associated with the accumulation of these aggregates. Apoptosis and autophagy are interconnected processes that can regulate each other in a balanced manner. The mechanism known as mTOR influences this balance. When mTOR is active, autophagy is upregulated. However, mTOR’s substrates regulate autophagy and multiple other biological processes. Inhibiting mTOR can affect crucial cellular functions like protein translation and ribosome biogenesis. Both autophagy and apoptosis are essential for maintaining cellular homeostasis. Autophagy removes misfolded proteins, damaged organelles, and long-lived cytosolic proteins, while apoptosis involves steps that prepare the targeted materials for transport to the lysosome. Various processes associated with apoptotic cell death lead to the formation of apoptotic bodies, which are then engulfed by macrophages or other cells [47,48,49,50]. The function of homeostasis depends on the balance of both phenomena. An imbalance between them can lead to neurodegenerative diseases like ALS, which are characterized by altered levels and activation in mTOR. The inhibition of autophagy decreases the cells’ ability to eliminate cellular waste. The abnormal and toxic protein aggregates worsen cellular stress in neurological disorders, resulting in cellular malfunction and disease.

Consequently, activating autophagy could help in restoring this pathologic state. Once the PI3K/Akt/mTOR complex is formed, the PI3K/AKT signaling (survival pathways) alters the activity of mTOR and Akt, activating autophagy-protecting neurons [48,51,52,53,54,55,56,57,58]. The deregulation of autophagy or apoptosis plays an essential role in the etiology and progression of numerous diseases. This imbalance has been associated with several neurodegenerative diseases, including ALS.

ALS and spinal muscular atrophy (SMA) share some of the exact pathophysiological mechanisms regarding synaptic plasticity and synaptogenesis, suggesting that mTOR may be involved [59,60,61,62,63]. The wild-type VAPB of the ALS8 protein functions at the endoplasmic reticulum (ER) membrane, regulating vesicular transport, maintaining homeostasis, and connecting the ER to the Golgi apparatus. In ALS8, on the other hand, overexpressing the mutant VAPBP56S causes wild-type VAPB to clump together and become active, which changes how the mitochondria-endoplasmic reticulum complex (MERC) works. The mTOR pathway activates these processes, including autophagy, apoptosis, and increased protein synthesis through C1q/TNF-related protein 3 (CTRP3) [62,63,64,65,66]. One possible explanation for motor neuron death is that when VAPB is mutated, mTOR cannot restore cellular homeostasis. The mechanism of mTOR is complex. Phosphorylated mTOR inhibits catabolism and enhances cell growth, whereas the non-phosphorylated form stimulates autophagy [35]. Autophagy activation through mTOR inhibition seems to be a promising therapeutic target for ALS and other neurodegenerative disorders. However, due to the toxicity of mTOR inhibitors, several studies have investigated potential substitutes for rapamycin. Additionally, mTOR impacts ALS and other neurodegenerative diseases. The complexity of mTOR’s metabolic signaling interactions currently prevents its use as a therapeutic target. Studies aimed at reducing the detrimental effects of mTOR inhibition in ALS and other neurodegenerative diseases require a thorough investigation of its mechanisms and the metabolic pathways involved in its signaling. Muscle atrophy, weakness, and paralysis often result from ubiquitin–proteasome, autophagy, and mitochondrial dysfunction, which are the main characteristics of ALS. These issues involve mTOR/AMPK signaling. ALS is a multifaceted neuromuscular disease characterized by muscle atrophy, motor neuron loss, and neuromuscular junction dysfunction. The majority of studies on ALS have focused on the central nervous system. However, recent approaches indicate how important peripheral tissues—especially skeletal muscle—are to disease onset and control, which may impact the neuromuscular junction (NMJ) and motor neurons, similarly to ALS-like muscle atrophy. An innovative strategy for ALS research could be utilized to investigate the interaction between muscles and neurons at the motor plate. Studies in other models have indicated that muscle protein release may influence neuron function [39,66]. A recently discovered protein, C1q/TNF-related protein 3 (CTRP3), assists muscles in regulating axonal local translation. This regulatory mechanism depends on the involvement of mTOR. Moreover, ALS has been linked to reduced axonal local protein synthesis [67]. A proposed pathophysiological connection exists between amyotrophic lateral sclerosis (ALS) and spinal muscular atrophy (SMA), indicating that these conditions may share fundamental underlying mechanisms [67]. Moreover, mTOR may significantly influence memory, synaptic plasticity, and the protein-mediated processes involved in synaptogenesis [67,68]. Reconceptualizing muscles as secretory organs presents an innovative and potential direction for ALS research [63,66]. Researchers have cautiously approached muscle as a secretory organ. According to these studies, the muscle protein CTRP-3 may impact the PI3K/AKT/mTOR signaling pathway and regulate the production of neuronal proteins. Decreased muscle CTRP-3 production may affect plaque neuron metabolism, leading to neuromuscular dysfunction. A protein like CTRP-3 can modify the neural plate’s function by inhibiting adjacent neurons’ neurotransmitter release, affecting muscle performance. Therefore, a thorough evaluation of the CRTP3-PI3K/Akt/mTOR axis in the context of muscular dysfunction in ALS is necessary.

## 4. Conclusions

The involvement of mTOR in ALS pathophysiology is highly suggestive, but studies are still in the early stages. The role of mTOR signaling in degenerative diseases such as ALS and other related conditions lies in its ability to inhibit autophagy, which prevents the elimination of misfolded proteins (cellular waste), contributing negatively to the disease. mTOR is associated with conditions that impact muscle function, including hypertrophy and atrophy. From this perspective, its role in ALS can be envisioned because muscle weakness and atrophy are the most common symptoms across the several types and subtypes of ALS and suggest a connection between mTOR and ALS. Previous studies in animal models of ALS indicate that dysfunction in skeletal muscle and the neuromuscular junction occurs before the degeneration of motor neurons and the onset of clinical symptoms. Evaluating the muscle as a secretory organ and how the protein CTRP-3 affects the PI3K/AKT/mTOR signaling pathway, which can control protein synthesis in neurons, could support the idea of neuromuscular dysfunction. Therefore, mTOR dysfunction in muscle secretion may influence the mTOR pathway and other signaling processes that regulate neurons and muscles, ultimately affecting the peripheral neuromotor function system. Nevertheless, this approach remains a hypothesis for potential discussion. Patients with ALS exhibit distinct features of muscle degeneration, while dementia is usually absent, which sets them apart from other neurodegenerative diseases. Therefore, emphasizing a peripheral focus rather than a purely neurocentric perspective may highlight the significance of mTOR signaling in ALS. Despite this, studies on ALS with a muscular focus are limited [66,67,69,70]. Crayle et al. [68] presented an intriguing proposition associating the lack of IGBP7 (insulin-like growth factor binding protein 7) expression with the activation of IGF-1, an mTOR activator. The authors associated IGBP7 absence with the reversal of clinical symptoms of ALS. Theoretically, it seems controversial because mTOR activation inhibits autophagy, which enhances the aggregation of protein and its deposition in the cytoplasm, which could exacerbate the disease. In truth, despite several studies and partial conclusions, we still know little or nothing definitive about ALS in all aspects of the disease, including onset and progression. mTOR and its metabolic connections may be a novel therapeutic resource, but this requires extensive study. The only certainty we have is that we know very little about ALS. With our current knowledge, we only promote an insignificant delay in the progression of the disease without any cure or control.

## Figures and Tables

**Figure 1 biomedicines-13-00952-f001:**
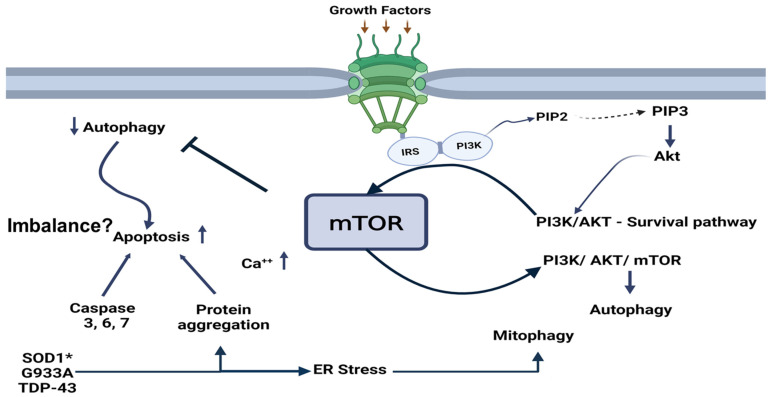
The model demonstrates metabolic signaling, illustrating how mTOR affects autophagy and apoptosis. The arrows mean activation; decreasing or increasing function for autophagy and apoptosis, respectively, while mTOR inhibits autophagy ( 
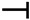
 ) IRS = insulin receptor substrate; PI3K = phosphatidylinositol (PI) 3-kinase; Akt = serine/th reonine kinase Akt, also known as protein kinase B (PKB); PIP2 = phosphatidylinositol biphosphate; PIP3 = phosphatidylinositol triphosphate.

**Figure 2 biomedicines-13-00952-f002:**
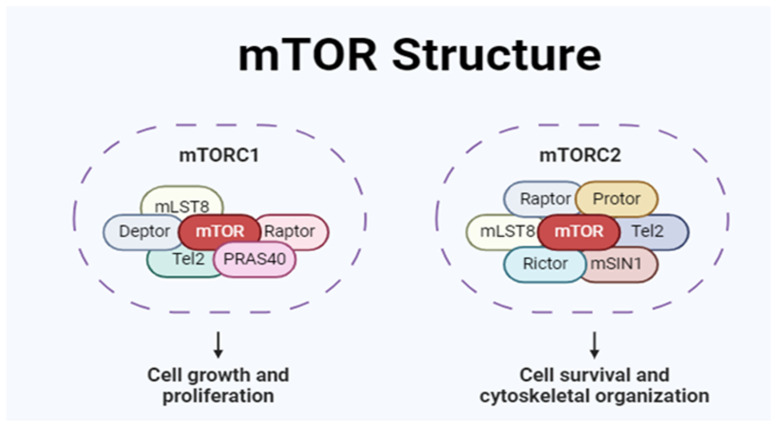
The mTOR pathway plays a crucial role in cellular regulation and consists of two main complexes: mTORC1 and mTORC2. mTORC1 is composed of proteins such as Deptor, PRAS40, GβL/mLST8, and the rapamycin-sensitive regulatory protein known as raptor. In contrast, mTORC2 includes GβL/mLST8, mSIN1, PRR5/Protor, Deptor, and Rictor. Understanding these complexes is essential for investigating their functions and potential therapeutic applications in cellular processes.

**Figure 3 biomedicines-13-00952-f003:**
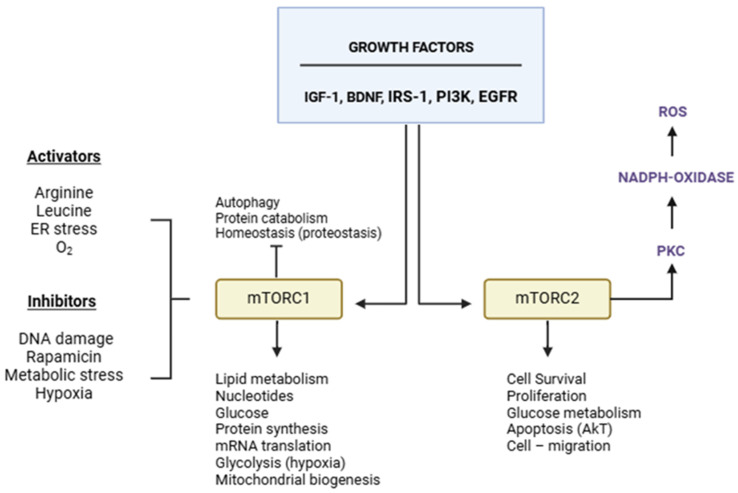
The functions of mTORC1 and mTORC2 are complex and involve various interactions with other signaling pathways, activation mechanisms, and self-regulatory processes. IRS-1 = insulin receptor substrate 1; IGF-1 = insulin growth factor 1; BDNF = brain-derived neurotrophic factor; PI3K = phosphoinositide 3-kinase; EGFR = epidermal growth factor receptor.

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
