# Peer review of "The Role of mTOR in Amyotrophic Lateral Sclerosis"

_biomedicines, 2025, doi:10.3390/biomedicines13040952_

Round 1

Reviewer 1 Report

Comments and Suggestions for Authors

In the abstract, the authors need to replace "energetic metabolism" with "energy metabolism" in line 15. Additionally, lines 21 to 25 lack clarity. To improve the abstract, it would be beneficial for the authors to align it with the background, aim, significance, and conclusion of the review. Clarifying these aspects would enhance the impact of the article and make the conclusion more engaging. This review needs extensive English editing service. 

  1. In the introduction section, lines 33 to 35 are unclear and confusing. Studies have well-documented that mutations in superoxide dismutase 1 (SOD1) result in protein misfolding and aggregation, leading to neurodegenerative amyotrophic lateral sclerosis (ALS). The authors need to rewrite these lines for better clarity. Additionally, the abbreviation "ALS" is repeatedly used in lines 43 to 46, even though it was already introduced in the abstract and the beginning of the introduction. The authors should avoid redundant abbreviations.

  2. The authors need to provide abbreviations for pathogen-associated molecular patterns (PAMPs) such as HMGB1 and RAGE when first mentioned in lines 52 and 53. These lines should also be rewritten for better clarity.

  3. Lines 57 to 64 require extensive rewriting, as they are not easily readable.

  4. The authors discuss mutations in SOD1 and TDP-43, which lead to muscle weakness or paralysis. However, while they mention the link between SOD1, oxidative stress, and ER stress, this is only depicted in Figure 1 and not explicitly discussed in the text. Additionally, the figure includes "growth factors" but does not specify which ones. The authors should provide specific names for the growth factors mentioned in the figure.

  5. The authors need to write detailed figure legends explaining the signaling pathways in Figure 1 and incorporate figure numbers within the text for better reference.

  6. The authors need to remove the BioRender signature from the bottom of Figures 1, 2, 3, and 4.

  7. The phrase "lab cells" in line 126 should be rephrased for better clarity. Additionally, the term "ALS8" in line 152 is unclear and needs further clarification. The phrase "enzymatic degradation" in line 155 also requires clarification.

  8. Reference 5 should be placed at the end of the "Concept of Evidence" section (line 174) instead of line 172.

  9. The phrase "endoplasmic reticulum-mediated cell death" in line 214 should be rephrased as "ER stress-mediated cell death."

  10. Lines 219 to 221 need to be rewritten for better clarity. Additionally, Figure 4 should be redrawn, and lines 294 to 297 require clarification.

  11. The conclusion should be concise and summarize the key findings of the review effectively. It requires extensive rewriting for better precision.

Comments on the Quality of English Language

This review needs extensive English editing service. 

Author Response

Referee 1:

Thank you for your suggestions. I am sure that they will significantly improve the review. All suggestions have been accepted and incorporated into the text, and the highlighted sections have been revised as suggested.

QUESTIONS

In the abstract, the authors need to replace "energetic metabolism" with "energy metabolism" in line 15. Additionally, lines 21 to 25 lack clarity.

Answer : mTOR  plays a pivotal role in controlling autophagy and affecting energy metabolism, in addition to supporting ...                                line 14

1.1 To improve the abstract, it would be beneficial for the authors to align it with the background, aim, significance, and conclusion of the review. Clarifying these aspects would enhance the impact of the article and make the conclusion more engaging. This review needs extensive English editing service. 

The Abstract was changed as suggested. Lines 09-24

Background: Amyotrophic lateral sclerosis (ALS) is a rare, progressive, and incurable disease characterized by muscle weakness and paralysis. Recent studies have explored a possible link between ALS pathophysiology and mTOR signaling. Recent reports have linked the accumulation of protein aggregates, dysfunctional mitochondria, and homeostasis to the development of ALS. mTOR plays a pivotal role in controlling autophagy and affecting energy metabolism, in addition to supporting neuronal growth, plasticity, and the balance between apoptosis and autophagy, all of which are important for homeostasis. Aim: This mini-review approaches the regulatory roles of mTOR signaling pathways, their interaction with other metabolic pathways, and their potential to modulate ALS progression. Significance: It discusses how these metabolic signaling pathways affect the neuromuscular junction, producing symptoms of muscle weakness and atrophy similar to those seen in patients with ALS. The discussion includes the concepts of neurocentric and peripheral and the possible connection between mTOR and neuromuscular dysfunction in ALS. Conclusion: It highlights the therapeutic potential of mTOR signaling and interconnections with other metabolic routes, making it a promising biomarker and therapeutic target for ALS.

­­­­­­­­­­­­­-----------------------------------------------------------------------------------------------------------------------------

  • In the introduction section, lines 33 to 35 are unclear and confusing. Studies have well-documented that mutations in superoxide dismutase 1 (SOD1) result in protein misfolding and aggregation, leading to neurodegenerative amyotrophic lateral sclerosis (ALS). The authors need to rewrite these lines for better clarity. Additionally, the abbreviation "ALS" is repeatedly used in lines 43 to 46, even though it was already introduced in the abstract and the beginning of the introduction. The authors should avoid redundant abbreviations.

The texts were clarified as suggested.

1.1 There are two types of ALS:  an idiopathic form known as sporadic ALS (sALS) and a genetically inherited type called familial ALS (fALS), which has over 30 identifiable subtypes of mutated genes. sALS develops epigenetic mutations that regulate the synthesis of mutated proteins, such as SOD1 and TARDBP [3,4]. -   lines  30-32.

1.2 Amyotrophic lateral sclerosis, one of the most significant motor neuron diseases, causes muscle degeneration and leads to paralysis [5]. The pathophysiological mechanisms of ALS are complex, involving intricate interactions among multiple factors and metabolic signaling. It is important to emphasize that frontotemporal dementia (FTD) shares significant pathological, genetic, and clinical similarities with ALS [6] - Lines 38-40

1.3 The redundant abbreviation was corrected.

2- The authors need to provide abbreviations for pathogen-associated molecular patterns (PAMPs) such as HMGB1 and RAGE when first mentioned in lines 52 and

 These lines should also be rewritten for better clarity.

The abbreviations were included and explained, as suggested.

  • Studies have been performed on the role of innate immunity in ALS. Its pathophysiology involves pathogen-associated molecular patterns (PAMPs) such as High mobility group box 1 protein (HMGB1), Receptor for advanced glycation end-products (RAGE), and Toll-like receptor 4 (TLR4) signaling axes, suggesting the perspective of the involvement of neuroinflammation in the disease [8] lines 43-46

3- Lines 57 to 64 require extensive rewriting, as they are not easily readable.

The text was modified as suggested.

Figure 1 shows hypothetical metabolic interactions involving mTOR. Several growth factors, including IGF-1 and GLP-1, activate mTOR through the phosphoinositide signaling cascade involving Akt (PKB) signaling. The PI3K/Akt pathway can also activate mTOR. The complexity of the process becomes clear when PI3K/Akt/mTOR signaling enhances autophagy, and the activation of mTOR by growth factors can inhibit it, and apoptosis is activated. This imbalance, mTOR-induced, increases misfolded protein aggregation and deposition in the cytoplasm and endoplasmic reticulum.

The process could involve mutated SOD1, changes in antioxidative mechanisms leading to oxidative stress, and endoplasmic reticulum stress with impaired mitochondrial function. These processes are observed in ALS and other neurodegenerative disorders. The main symptom of ALS is progressive neuromuscular atrophy, and there is a proposed link between the dysregulation of IGF-1/GLP-1 signaling and ALS. By inhibiting the PI3K/Akt/mTOR and MAPK/ERK pathways, IGF-1 and GLP-1 prevent neuronal death resulting from amyloidogenesis, cerebral glucose deprivation, and neuroinflammation. Neurodegenerative disease progression accelerates in patients with IGF-1 resistance and GLP-1 deficiency [9–13]. The complexity of the process becomes evident when PI3K/Akt/mTOR signaling stimulates autophagy (Figure 1). At the same time, growth factors also activate mTOR, which subsequently inhibits it, and this inhibition leads to the activation of apoptosis. Despite the extensive literature on this topic, the relationship between signaling pathways related to mTOR activation and homeostasis remains unclear. Understanding these metabolic interactions in ALS is particularly complex. All interaction studies are open for discussion and study, including those on healthy individuals and those with specific pathologies.   lines 49-65

  • The authors discuss mutations in SOD1 and TDP-43, which lead to muscle weakness or paralysis. However, while they mention the link between SOD1, oxidative stress, and ER stress, this is only depicted in Figure 1 and not explicitly discussed in the text. Additionally, the figure includes "growth factors" but does not specify which ones. The authors should provide specific names for the growth factors mentioned in the figure.

 Growth factors were included in the explanation for Figure 1, as suggested. Part of the response to question 4 can be found in the text provided in answer 3 (lines 49-65). Abbreviations are included in the legend of Figure 1. The examples of mTOR activators are shown in detail in Figure 2

5-The authors need to write detailed figure legends explaining the signaling pathways in Figure 1 and incorporate figure numbers within the text for better reference.

The legend- figure 1 was completed, and its explanation was included in the text (lines 49-65)

Figure 1. The model for metabolic signaling illustrates how mTOR influences autophagy and apoptosis in ALS (amyotrophic lateral sclerosis).            = activation;           = inhibition.  Sometimes, this means    decreasing or    increasing function for autophagy and apoptosis, respectively. IRS = Insulin receptor substrate; PI3K = phosphatidylinositol (PI) 3-kinase; Akt = serine/threonine kinase Akt, also known as protein kinase B (PKB); PIP2 = phosphatidylinositol biphosphate; PIP3 = phosphatidylinositol triphosphate.

  • The authors need to remove the BioRender signature from the bottom of Figures 1, 2, 3,

         BioRender was omitted in all the Figures (1-3), as suggested.

  • The phrase "lab cells" in line 126 should be rephrased for better clarity. Additionally, the term "ALS8" in line 152 is unclear and needs further clarification. The phrase "enzymatic degradation" in line 155 also requires clarification.

7.1 The phrase has been rewritten as follows:

Line 126 : Lab cells were omitted. It has no sense in the phase which was changed to :

the term "ALS8" in line 152

Patients with familial ALS type 8 (ALS8) show elevated intracellular mTOR levels when compared to healthy controls. This increase in intracellular mTOR is suggested to be due to a possible dysfunction in the exocytosis process [41]. This finding has only been observed in cells from ALS8 patients and cannot be generalized to all types of familial ALS. The high intracellular mTOR content can inhibit autophagy and increase the presence of misfolded and aggregated proteins in the cytoplasm, as well as dysregulate the endoplasmic reticulum and alter enzymatic function, all of which could significantly affect both sporadic ALS and familial ALS. Lines 136-141.

  • Reference 5 should be placed at the end of the "Concept of Evidence" section (line 174) instead of line 172.

The reference Granatiero et al. [5] was suggested to be moved from lines 172 to 174. However, it is already on line 153, and in the modified version, it appears on line 158.

Granatiero et al. [5] reported that in astrocytes with SOD1 mutations, the activation of mTOR is associated with increased expression of the insulin-like growth factor 1 receptor (IGF1R). Elevated levels of IGF1R activate the mTOR pathway and subsequently lead to the inhibition of autophagy. Interestingly, inhibiting the IGF1R-mTOR pathway has been shown to mitigate the harmful effects on motor neurons in the astrocytes of SOD1 mutant mice, indicating a potential connection between mTOR activity and motor neuron damage in ALS, highlighting mTOR's complex and nuanced role [5].

  • The phrase "endoplasmic reticulum-mediated cell death" in line 214 should be rephrased as "ER stress-mediated cell death."

     The modification was included as suggested: line 184

  1. Other signs observed in both ALS and FTD include the formation of misfolded TDP-43 and altered SOD1 or FUS. Furthermore, mitochondrial dysfunction, oxidative stress, protein imbalance, altered autophagy, ER stress-mediated cell death, stress granule formation, and altered RNA processing accelerate the disease's progression [50–52]. Lines 183-185

  • Lines 219 to 221 need to be rewritten for better clarity. Additionally, Figure 4 should be redrawn, and lines 294 to 297 require clarification.

Multiple mutations in the TARDBP gene have been associated with ALS [52,53]. As a result, abnormal functions of TDP-43, including protein aggregation, relocation from the nucleus to the cytoplasm, and the accumulation of ubiquitinated and hyperphosphorylated TDP-43 in inclusion bodies, play a pivotal role in the disease's pathophysiology. Lines 187-190

Figure 4 is not very effective, and the explanation provided in the text seems sufficient. Therefore, Figure 4 will be omitted from the review.

11- The conclusion should be concise and summarize the key findings of the review effectively. It requires extensive rewriting for better precision.

The conclusion was rewritten :

  1. Conclusion

The involvement of mTOR in ALS pathophysiology is highly suggestive, but studies are still in the early stages. The role of mTOR signaling in degenerative diseases such as ALS and other related conditions lies in its ability to inhibit autophagy, which prevents the elimination of misfolded proteins (cellular waste), contributing negatively to the disease.   mTOR is associated with conditions that impact muscle function, including hypertrophy and atrophy. From this perspective, its role in ALS can be envisioned because muscle weakness and atrophy are the most com-mon symptoms across the several types and subtypes of ALS and suggest a connection between mTOR and ALS. Previous studies in animal models of ALS indicate that dysfunction in skeletal muscle and the neuromuscular junction occurs before the degeneration of motor neurons and the onset of clinical symptoms. Evaluating the muscle as a secretory organ and how the protein CTRP-3 affects the PI3K/AKT/mTOR signaling pathway, which can control protein synthesis in neurons, could support the idea of neuromuscular dysfunction. Therefore, mTOR dys-function in muscle secretion may influence the mTOR pathway and other signaling processes that regulate neurons and muscles, ultimately affecting the peripheral neuromotor function system. Nevertheless, this approach remains a hypothesis for potential discussion. Patients with ALS exhibit distinct features of muscle degeneration, while dementia is usually absent, which sets them apart from other neurodegenerative diseases. Therefore, emphasizing a peripheral focus rather than a purely neurocentric perspective may highlight the significance of mTOR signaling in ALS. Despite this, studies on ALS with muscular focus are limited (77, 78). Crayle et al. [79] presented an intriguing proposition associating the lack of IGBP7 (insulin-like growth factor binding protein 7) expression with the activation of IGF-1, an mTOR activator. The authors associated IGBP7 absence with the reversal of clinical symptoms of ALS. Theoretically, it seems controversial because mTOR activation inhibits autophagy, which enhances the aggregation of protein and its deposition in the cytoplasm, which could exacerbate the disease. In truth, despite several studies and partial conclusions, we know little or still nothing definitive about ALS in all aspects of the disease, including onset and progression. mTOR and its metabolic connections may be a novel therapeutic resource, but this requires extensive study. The only certainty we have is that we know very little about ALS. With our current knowledge, we only promote an insignificant delay in the progression of the disease without any cure or control.

I reread the text and tried to identify the points you raised. I found and corrected some, but I also removed others from the text because, as you mentioned, I felt they were unnecessary. A native English speaker has reviewed the manuscript, and all suggestions have been incorporated. I appreciate your comments.

Referee 2

Thank you for your suggestions. I am sure that they will significantly improve the review. All suggestions have been accepted and incorporated into the text, and the highlighted sections have been revised as suggested.

  • The paper explores the relationship between mTOR and ALS, which shows some innovation. However, further clarification on the feasibility and effectiveness of mTOR as a therapeutic target would more effectively drive the development of ALS treatment.

 I agree with your observation. However, we are still far from having a definitive proposal for mTOR as a therapeutic target or biomarker. The current suggestion is based on mTOR's function and the common muscular symptoms observed in all types of ALS. While mTOR is a promising indicator, we must thoroughly understand its signaling and the metabolic connections involved before considering it as a definitive therapeutic resource. The aim is to open the discussion on mTOR and initiate new studies. Would it be premature to propose it as a definitive therapeutic approach? I believe so. I have modified my conclusions for this reason. I conclude, "In truth, despite numerous studies and partial findings, we still know little or nothing definitive about ALS in all aspects, including onset and progression. mTOR and its metabolic connections could represent a novel therapeutic resource that requires extensive studies. I hope you share my doubts.

  • When discussing the effects of mTORC1 and mTORC2 on Akt, only the opposite actions are stated briefly, without elaborating on the synergistic or antagonistic mechanisms in the pathological process of ALS. New concepts like the impact of muscle secretion on nerve function are only introduced superficially, lacking in-depth analysis and evidence.

It is a very interesting question, but answering it definitively is challenging.

Figures 1, 2, and 3 schematically summarize the functions of mTORC1 and mTORC2 and their functional associations. The mechanisms behind their actions are largely unknown. Growth factors, including IGF-1 and EGFR, activate both pathways. Due to the incomplete understanding of each metabolic connection, pinpointing the exact moment their actions converge or diverge is challenging. Crayle's suggestion highlights this complexity.

"Crayle et al. [79] presented an intriguing proposition associating the lack of IGBP7 (insulin-like growth factor binding protein 7) expression with the activation of IGF-1, an mTOR activator. The authors associated IGBP7 absence with the reversal of clinical symptoms of ALS. Theoretically, it seems controversial because mTOR activation inhibits autophagy, which enhances the aggregation of protein and its deposition in the cytoplasm, which could exacerbate the disease.

mTORC1 inhibits autophagy, while mTORC2 activates apoptosis. Both autophagy and apoptosis are balanced to maintain homeostasis in non-pathological conditions. An imbalance that favors apoptosis due to the inhibition of autophagy could be detrimental to cells, as accumulated cellular waste, including aggregated protein deposits, cannot be digested by lysosomes. Figure 3 summarizes some functions of mTORC1 and mTORC2.

The discussion about muscular secretion is extensive, as it will determine whether the neurodegenerative disease ALS is exclusively neurocentric, peripheral, or a combination of both. This discussion is still in its early stages, and numerous studies are required to clarify this. Thus, the discussion was only introduced because there is no definitive conclusion. 

  • The logical coherence within some paragraphs is insufficient. When discussing the relationship between mTOR and TDP - 43, the description jumps from the abnormal changes of TDP - 43 in ALS to the influence of mTOR on it, lacking necessary logical connections and making it difficult for readers to understand.

I suppose you refer to lines 20-15 (original). I agree with you; a link was rewritten to induce a logical connection, and another part was omitted to clarify the text.

Reports indicate that findings on mTOR inhibition are inconsistent, suggesting that results may change based on the type of ALS, specific gene mutations, and the metabolic signaling involved. In SOD1 mutants, mTOR inhibition worsens ALS, while similar mTOR inhibition in elevated TDP-43 levels protects against neuronal degeneration [43-45] despite the pathogenesis of ALS being associated with TDP-43 [47–49].

 Lines 177-180

  • The language expression is generally smooth, but there are some inaccuracies in the use of professional terms. When describing processes related to the mTOR signalling pathway, some terms are vaguely expressed, which may cause ambiguity. Some sentences are also a bit wordy, and some paragraphs could be streamlined to improve readability.

I reread the text and tried to identify the points you raised. I found and corrected some, but I also removed others from the text because, as you mentioned, I felt they were unnecessary. A native English speaker has reviewed the manuscript, and all suggestions have been incorporated. I appreciate your comments.

Reviewer 2 Report

Comments and Suggestions for Authors

The paper explores the relationship between mTOR and ALS, which shows some innovation. However, further clarification on the feasibility and effectiveness of mTOR as a therapeutic target would more effectively drive the development of ALS treatment.

When discussing the effects of mTORC1 and mTORC2 on Akt, only the opposite actions are stated briefly, without elaborating on the synergistic or antagonistic mechanisms in the pathological process of ALS. New concepts like the impact of muscle secretion on nerve function are only introduced superficially, lacking in - depth analysis and evidence.

The logical coherence within some paragraphs is insufficient. When discussing the relationship between mTOR and TDP - 43, the description jumps from the abnormal changes of TDP - 43 in ALS to the influence of mTOR on it, lacking necessary logical connections and making it difficult for readers to understand.

The language expression is generally smooth, but there are some inaccuracies in the use of professional terms. When describing processes related to the mTOR signalling pathway, some terms are vaguely expressed, which may cause ambiguity. Some sentences are also a bit wordy, and some paragraphs could be streamlined to improve readability.

Comments on the Quality of English Language

The language expression is generally smooth, but there are some inaccuracies in the use of professional terms. When describing processes related to the mTOR signalling pathway, some terms are vaguely expressed, which may cause ambiguity. Some sentences are also a bit wordy, and some paragraphs could be streamlined to improve readability.

Author Response

Referee 2

Thank you for your suggestions. I am sure that they will significantly improve the review. All suggestions have been accepted and incorporated into the text, and the highlighted sections have been revised as suggested.

  • The paper explores the relationship between mTOR and ALS, which shows some innovation. However, further clarification on the feasibility and effectiveness of mTOR as a therapeutic target would more effectively drive the development of ALS treatment.

I agree with your observation. However, we are still far from having a definitive proposal for mTOR as a therapeutic target or biomarker. The current suggestion is based on mTOR's function and the common muscular symptoms observed in all types of ALS. While mTOR is a promising indicator, we must thoroughly understand its signaling and the metabolic connections involved before considering it as a definitive therapeutic resource. The aim is to open the discussion on mTOR and initiate new studies. Would it be premature to propose it as a definitive therapeutic approach? I believe so. I have modified my conclusions for this reason. I conclude, "In truth, despite numerous studies and partial findings, we still know little or nothing definitive about ALS in all aspects, including onset and progression. mTOR and its metabolic connections could represent a novel therapeutic resource that requires extensive studies. I hope you share my doubts.

  • When discussing the effects of mTORC1 and mTORC2 on Akt, only the opposite actions are stated briefly, without elaborating on the synergistic or antagonistic mechanisms in the pathological process of ALS. New concepts like the impact of muscle secretion on nerve function are only introduced superficially, lacking in-depth analysis and evidence.

It is a very interesting question, but answering it definitively is challenging.

Figures 1, 2, and 3 schematically summarize the functions of mTORC1 and mTORC2 and their functional associations. The mechanisms behind their actions are largely unknown. Growth factors, including IGF-1 and EGFR, activate both pathways. Due to the incomplete understanding of each metabolic connection, pinpointing the exact moment their actions converge or diverge is challenging. Crayle's suggestion highlights this complexity.

"Crayle et al. [79] presented an intriguing proposition associating the lack of IGBP7 (insulin-like growth factor binding protein 7) expression with the activation of IGF-1, an mTOR activator. The authors associated IGBP7 absence with the reversal of clinical symptoms of ALS. Theoretically, it seems controversial because mTOR activation inhibits autophagy, which enhances the aggregation of protein and its deposition in the cytoplasm, which could exacerbate the disease."

mTORC1 inhibits autophagy, while mTORC2 activates apoptosis. Both autophagy and apoptosis are balanced to maintain homeostasis in non-pathological conditions. An imbalance that favors apoptosis due to the inhibition of autophagy could be detrimental to cells, as accumulated cellular waste, including aggregated protein deposits, cannot be digested by lysosomes. Figure 3 summarizes some functions of mTORC1 and mTORC2.

The discussion about muscular secretion is extensive, as it will determine whether the neurodegenerative disease ALS is exclusively neurocentric, peripheral, or a combination of both. This discussion is still in its early stages, and numerous studies are required to clarify this. Thus, the discussion was only introduced because there is no definitive conclusion.

  • The logical coherence within some paragraphs is insufficient. When discussing the relationship between mTOR and TDP - 43, the description jumps from the abnormal changes of TDP - 43 in ALS to the influence of mTOR on it, lacking necessary logical connections and making it difficult for readers to understand.

I suppose you refer to lines 20-15 (original). I agree with you; a link was rewritten to induce a logical connection, and another part was omitted to clarify the text.

Reports indicate that findings on mTOR inhibition are inconsistent, suggesting that results may change based on the type of ALS, specific gene mutations, and the metabolic signaling involved. In SOD1 mutants, mTOR inhibition worsens ALS, while similar mTOR inhibition in elevated TDP-43 levels protects against neuronal degeneration [43-45] despite the pathogenesis of ALS being associated with TDP-43 [47–49].

Lines 177-180

  • The language expression is generally smooth, but there are some inaccuracies in the use of professional terms. When describing processes related to the mTOR signalling pathway, some terms are vaguely expressed, which may cause ambiguity. Some sentences are also a bit wordy, and some paragraphs could be streamlined to improve readability.

I reread the text and tried to identify the points you raised. I found and corrected some, but I also removed others from the text because, as you mentioned, I felt they were unnecessary. A native English speaker has reviewed the manuscript, and all suggestions have been incorporated. I appreciate your comments.

Round 2

Reviewer 1 Report

Comments and Suggestions for Authors

The authors made the corrections, and the manuscript was improved for the readers.

Author Response

Thank you for your suggestions.

Reviewer 2 Report

Comments and Suggestions for Authors
  • Ensure all figures are cited in numerical order and described clearly.
  • The relationship between mTORC1/mTORC2 dysregulation and specific ALS subtypes (e.g., SOD1, TDP-43) is not fully clarified.
  • Multiple citation errors exist (e.g., author names misspelled, journal abbreviations inconsistent).
  • Expand on how mTOR dysregulation specifically impacts ALS subtypes (e.g., SOD1 mutants vs. TDP-43opathies).
  • Provide more context for CTRP3’s role in ALS, referencing recent studies if available.
Comments on the Quality of English Language

The English could be improved to more clearly express the research.

Author Response

I appreciate the second round of questions. You have provided several insightful observations, and although some points were clarified in the text, others were not answered directly but were justified instead.

The Role of mTOR in Amyotrophic Lateral Sclerosis

Comments and Suggestions for Authors

  • Ensure all figures are cited in numerical order and described clearly.

 The figures referenced in the text were conferred.

  • The relationship between mTORC1/mTORC2 dysregulation and specific ALS subtypes (e.g.,

SOD1, TDP-43) is not fully clarified.

To answer your questions directly, we must first understand how mTOR signaling interacts with mutated SOD1 and TDP-43 metabolic signaling. A few and isolated manuscripts examine this but do not suggest any possible metabolic signaling interaction, leaving this issue without an effective response.

The potential functional relationship between mTORC1 and mTORC2 is intricate. Both complexes are mainly activated by growth factors such as GLP-1 and IGF-1. Additionally, leucine, oxygen, insulin, and energy levels can activate mTORC1, while stress and the drug rapamycin can inhibit it. The mTORC1 complex plays a crucial role in downregulating autophagy, which impacts cellular homeostasis. However, it promotes the biosynthesis of macromolecules and activates several metabolic signaling pathways.

In contrast, mTORC2 is involved in cell survival and cytoskeletal organization. It is important to note that mTORC1 and mTORC2 can have different effects depending on the tissue type. Figure 3 shows part of these interactions. The pathological conditions involving mTOR can be due to its dysfunction or may be linked to changes in metabolic signaling that mTOR induced or regulated. In reality, we still have much to learn about the complete mechanisms involved in the relationship between mTOR and muscle weakness to understand what happens in ALS. Since muscle weakness is a common symptom observed in all forms of ALS, studies on mTOR are significant. It is known that physical exercise, nutrients, or insulin via Akt signaling can activate mTORC1 in muscle tissue. However, the proposition of mTOR as a potential biomarker or therapeutic target for ALS is relatively novel. The reports linking ALS and mTOR are few, and most of them approach autophagy and the use of chaperones and nothing about muscular weakness.  The dysregulation of mTOR related to a specific fALS subtype is unclear, and there is little literature about it.  For example, SOD1 is a significant mutation for sALS pathology, but this mutation does not occur in all forms of fALS, as already reported for fALS8.  The TDP-43 protein and mutations in the C9orf72, SOD1, and FUS genes have been studied in sALS. It is important to note that no literature exists on these mutations associated with  mTOR signaling.   Few studies have concentrated on specific subtypes of fALS. FALS is rare.  

  • Multiple citation errors exist (e.g., author names misspelled, journal abbreviations

inconsistent).

 The references were evaluated using Mendeley software, and some errors were corrected. 

  • Expand on how mTOR dysregulation specifically impacts ALS subtypes (e.g., SOD1 mutants
  1. TDP-43opathies).

Amyotrophic lateral sclerosis (ALS) encompasses a spectrum of incurable neurodegenerative diseases characterized by similar clinical symptoms. Muscle weakness is predominant, and usually, there is paralysis of the upper and lower limbs, progressing to respiratory muscle paralysis. Ninety percent of the cases are sporadic ALS cases.”

Most studies focus on sporadic ALS. There are few reports on each of the over 30 subtypes of fALS.  It is essential to recognize that each subtype exhibits different gene mutations and consequences a predominant metabolic network. For example, the SOD1 mutation is present in sALS and absent in patients with fALS8.  Thus, the available knowledge is predominantly about sALS.  We aimed to connect the common symptom of muscle weakness observed in ALS with mTOR signaling because of its recognized role in hypertrophy, hypotrophy, and muscle dysfunction.  Currently, literature is limited in focusing on each subtype of fALS (over 30).

  • Provide more context for CTRP3’s role in ALS, referencing recent studies if available.

The explanation of the innovative role of CTRP3 was included in the text :  lines 237-244. 

Round 3

Reviewer 2 Report

Comments and Suggestions for Authors

No other suggestions.

Comments on the Quality of English Language

No other suggestions.

Author Response

We would like to thank reviewer for all the insightful observations.
